# Nelfinavir Inhibits the TCF11/Nrf1-Mediated Proteasome Recovery Pathway in Multiple Myeloma

**DOI:** 10.3390/cancers12051065

**Published:** 2020-04-25

**Authors:** Dominika Fassmannová, František Sedlák, Jindřich Sedláček, Ivan Špička, Klára Grantz Šašková

**Affiliations:** 1Institute of Organic Chemistry and Biochemistry of the Czech Academy of Sciences, Flemingovo n. 2, 16610 Prague, Czech Republic; 2Department of Genetics and Microbiology, Charles University, Viničná 5, 12843 Prague, Czech Republic; 3First Faculty of Medicine, Charles University, Kateřinská 32, 12108 Prague, Czech Republic; 41st Department Medicine—Department of Hematology, Charles University, U Nemocnice 2, 12808 Prague, Czech Republic

**Keywords:** proteasome, TCF11/Nrf1, DDI2, nelfinavir, multiple myeloma, proteasome inhibitors, bortezomib, carfilzomib

## Abstract

Proteasome inhibitors are the backbone of multiple myeloma therapy. However, disease progression or early relapse occur due to development of resistance to the therapy. One important cause of resistance to proteasome inhibition is the so-called bounce-back response, a recovery pathway driven by the TCF11/Nrf1 transcription factor, which activates proteasome gene re-synthesis upon impairment of the proteasome function. Thus, inhibiting this recovery pathway potentiates the cytotoxic effect of proteasome inhibitors and could benefit treatment outcomes. DDI2 protease, the 3D structure of which resembles the HIV protease, serves as the key player in TCF11/Nrf1 activation. Previous work found that some HIV protease inhibitors block DDI2 in cell-based experiments. Nelfinavir, an oral anti-HIV drug, inhibits the proteasome and/or pAKT pathway and has shown promise for treatment of relapsed/refractory multiple myeloma. Here, we describe how nelfinavir inhibits the TCF11/Nrf1-driven recovery pathway by a dual mode of action. Nelfinavir decreases the total protein level of TCF11/Nrf1 and inhibits TCF11/Nrf1 proteolytic processing, likely by interfering with the DDI2 protease, and therefore reduces the TCF11/Nrf1 protein level in the nucleus. We propose an overall mechanism that explains nelfinavir’s effectiveness in the treatment of multiple myeloma.

## 1. Introduction

The ubiquitin-proteasome system (UPS) is the key regulator in the maintenance of cellular proteostasis. The 26S proteasome, an ATP-dependent multi-protease complex, plays the central role by degrading unneeded proteins that have been polyubiquitinated by an enzymatic cascade termed E1, E2s, and E3s. Dysregulation of the sensitive balance between protein synthesis and degradation often results in severe disorders, such as cancer. Although the survival of cancer cells is often dependent on the UPS, cancer cells are mostly resistant to proteasome inhibitors. Multiple myeloma (MM) is an exception. MM cells are sensitive to proteasome inhibitors due to excessive activation of the unfolded protein response (UPR) [1,2], and proteasome inhibitors form the backbone of MM therapy [3,4]. Nevertheless, MM cells sometimes develop early resistance to proteasome inhibitor treatment.

One important cause of resistance to proteasome inhibition is the so-called bounce-back response, in which proteasome activity is restored by TCF11/Nrf1-driven proteasome re-synthesis [5,6] (Figure 1A). TCF11/Nrf1 is encoded by the NFE2L1 gene and belongs to the CNC-bZIP (Cap’n’Collar/basic leucine zipper) family of transcription factors. It is ubiquitously expressed in human tissues and resides in the endoplasmic reticulum (ER) membrane under normal conditions. To maintain low levels, it is coupled to ER-associated protein degradation (ERAD) requiring E3-ligase HRD1 and AAA-ATPase p97/VCP [7]. Upon cell exposure to proteotoxic stress by proteasome inhibition or oxidants, TCF11/Nrf1 detaches from the ER membrane, activates, and translocates to the nucleus, where it binds to antioxidant response elements (AREs) in the promotor regions of almost all proteasome genes and other UPS-related genes and activates their expression [5,6,8,9]. Importantly, proteolytic cleavage in the cytosol by the DDI2 protease is required for TCF11/Nrf1 activation, although a detailed mechanism of this cleavage has not yet been described [10,11]. DDI2 contains an N-terminal ubiquitin-like domain, C-terminal ubiquitin-interacting motif, and a helical domain that precedes the aspartic protease. The 3D structure of DDI2 strikingly resembles that of the HIV-1 protease (Figure 1C) [12], and therefore, HIV-1 protease inhibitors (HIV PIs) are the first choice to inhibit DDI2 activity. Although direct binding of HIV PIs to DDI2 in vitro has not been explicitly confirmed [12], some HIV PIs showed significant in vivo activity against yeast Ddi1 [13]. Moreover, a yeast Ddi1 knockout strain complemented with the human DDI1 ortholog exhibited 97% growth inhibition when subjected to 25 μM nelfinavir, and 90% growth inhibition when subjected to 25 μM tipranavir, with IC_50_ values of 3.4 ± 0.4 μM and 1.5 ± 0.2 μM, respectively [13].

Nelfinavir is the first-generation HIV PI that appears active against solid tumors, leukemia, and MM both in vitro and in vivo in preclinical experiments [14,15,16]. Moreover, nelfinavir showed promise for overcoming proteasome inhibitor resistance in MM in a phase I clinical trial [17] and for treatment of the proteasome inhibitor-refractory MM in a phase 2 clinical trial [18]. The molecular mechanisms underlying nelfinavir’s efficacy for human malignancies are still under investigation, but cell-based studies have provided some clues. Nelfinavir triggers the UPR by inhibiting UPR-activating protease S2P, resulting in apoptosis of murine liposarcoma [19,20]. Furthermore, nelfinavir interferes with the pAKT pathway [21,22,23] and inhibits proteasome activity [24,25].

Here, we bring new insights into how nelfinavir affects the growth of MM cells. First, we demonstrate that downregulation of DDI2 in MM cells leads to a significant decrease in proteasome gene expression. We further identify nelfinavir as the most potent HIV PI that inhibits proteasome recovery. Nelfinavir accomplishes this by affecting TCF11/Nrf1-mediated proteasome re-synthesis by a dual mode of action. In a detailed analysis, we show that nelfinavir inhibits proteolytic processing of transcription factor TCF11/Nrf1, suggesting that it might interfere with DDI2 activity.

## 2. Results

### 2.1. Efficient Proteasome Re-Synthesis in MM Is Dependent on DDI2 and Can Be Attenuated by HIV PI Nelfinavir

TCF11/Nrf1-mediated proteasome re-synthesis (see the scheme in Figure 1A) is an attractive target for therapeutic intervention, especially for MM and mantle cell lymphoma, which are currently treated with proteasome inhibitors [5,11]. When proteasome function is impaired, DDI2 protease cleaves and activates TCF11/Nrf1 [10], which in turn upregulates proteasome gene expression. Downregulation or impairment of DDI2 leads to decreased levels of activated Nrf1 in NIH-3T3 and HCT116 cells [10,26]. To assess whether DDI2 impairment leads to downregulation of proteasome gene expression in MM cells, we used a CRISPRi method targeting DDI2 in the RPMI8226 cell line. Cells were treated with lentiviral particles harboring DDI2 CRISPRi alone or in combination with 20 nM bortezomib (BTZ), a clinically used proteasome inhibitor. RT-qPCR analysis showed a significant decrease in three of TCF11/Nrf1’s target proteasome subunits (PSMB7, PSMC4, and PSMD12) when cells were co-treated with DDI2 CRISPRi lentiviral particles and BTZ (DDI2 CRISPRi + BTZ) compared to the control treated with mock lentiviral particles and BTZ (BTZ + mock) (Figure 1B).

As DDI2 is an aspartyl protease with a 3D structure strikingly similar to that of the HIV protease [12] (Figure 1C), we next aimed to test whether HIV PIs used in clinical practice affect TCF11/Nrf1-mediated proteasome re-synthesis. We generated a HEK293 cell line stably expressing firefly luciferase under 3xPSMA-ARE responsive elements where the TCF11/Nrf1 transcription factor binds, as previously described [5]. These cells were co-treated with MG132 and HIV PIs, and the luciferase activity was measured after 16 h. In parallel, we also generated the control U2OS cell line stably expressing a short-lived Green Fluorescent Protein (Ub^G76V^-GFP, GFP-degron) reporter to monitor proteasome recovery [27]. The U2OS Ub^G76V^-GFP cells were treated with covalent proteasome inhibitor carfilzomib (CFZ) for 2 h. After CFZ washout, the cells were treated with HIV PIs. After 20 h, GFP fluorescence was measured as a function of recovery of proteasome activity. In both assays, nelfinavir showed the most potent activity by inhibiting the TCF11/Nrf1-pathway and proteasome activity. Tipranavir displayed the second most potent inhibition (Figure 1D).

### 2.2. Nelfinavir Inhibits the TCF11/Nrf1 Pathway by a Dual Mode of Action and Decreases the TCF11/Nrf1 Levels in the Nucleus

To determine the half-maximal inhibitory and effective concentrations of nelfinavir in both assays, we treated the cell lines with increasing, low micromolar concentrations of nelfinavir (1 μM to 20 μM). Interestingly, the dose–response curve generated from the luciferase assay, which reflects inhibition of the TCF11/Nrf1 pathway by nelfinavir, was better fit by a bi-sigmoidal curve than a sigmoidal curve. The IC_50_ of nelfinavir was 6.8 ± 2.4 μM from a sigmoidal fit, while IC_50_ (1) of 3.6 ± 2.4 μM and IC_50_(2) of 15.8 ± 1.8 μM were derived from the bi-sigmoidal fit (Figure 2A, left panel). The GFP-degron assay revealed an even more pronounced bi-sigmoidal dose–response effect; sigmoidal fitting was impossible (Figure 2A, right panel). The half-maximal effective concentrations of nelfinavir determined with this short-lived GFP assay were 1.5 ± 0.7 μM and 14.9 ± 1.0 μM. These data suggest that nelfinavir at low micromolar concentrations exhibits two modes of action.

Next, we explored whether nelfinavir affects TCF11/Nrf1 processing to the nucleus. We treated HEK293 cells with 200 nM BTZ alone or in combination with 10 μM or 20 μM nelfinavir for 16 h and analyzed the amount of processed TCF11/Nrf1 transcription factor in the nuclei by high-throughput confocal microscopy (Figure 2B). Cells co-treated with BTZ and nelfinavir had significantly lower levels of TCF11/Nrf1 in the nuclei compared to the BTZ-treated cells. Interestingly, nelfinavir alone decreased the TCF11/Nrf1 level in the nucleus to a level comparable to DMSO treatment.

### 2.3. Nelfinavir Inhibits Proteolytic Processing of TCF11/Nrf1 in MM Cells

To further investigate the mechanism of action by which nelfinavir inhibits the TCF11/Nrf1 pathway, we tested whether nelfinavir interferes with TCF11/Nrf1 proteolytic processing. HEK293 cells and MM cell lines OPM-2 and RPMI8226 were treated with BTZ or nelfinavir alone and in combination. The cell lysates were then subjected to Western blotting, and the unprocessed (ER-embedded) and processed (ER-cleaved) forms of TCF11/Nrf1 were quantified (Figure 3, Appendix A). In HEK293 cells, which are less sensitive to proteasome inhibitor treatment, the total expression of TCF11/Nrf1 significantly decreased upon nelfinavir treatment. When TCF11/Nrf1 activation was induced by BTZ, the total amount of TCF11/Nrf1 decreased upon nelfinavir co-treatment. This effect was even more pronounced in the myeloma cell lines. Importantly, nelfinavir decreased the amount of the proteolytically processed form of TCF11/Nrf1 in both HEK293 cells and MM cell lines. The data suggest that the two modes of action of nelfinavir identified by the reporter assays are reflected by (1) decreased TCF11/Nrf1 expression and (2) inhibited TCF11/Nrf1 proteolytic processing.

### 2.4. Nelfinavir Decreases TCF11/Nrf1 Gene Expression and Activates Nrf2

Next, we explored the effect of nelfinavir on downstream TCF11/Nrf1 signaling. We performed RT-qPCR analysis of HEK293 cells and OPM-2 and RPMI8226 MM cell lines treated with BTZ and nelfinavir alone or in combination (Figure 4). First, we analyzed the levels of mRNA expression of TCF11/Nrf1’s target proteasome subunits (PSMB7, PSMC4, and PSMD12). In HEK293 cells, nelfinavir alone induced proteasome gene upregulation. While the OPM-2 cell line appeared sensitive to nelfinavir treatment, proteasome genes were unaffected in RPMI8226 cells. Surprisingly, in all three cell lines, the proteasome genes were unaffected when cells were co-treated with BTZ and nelfinavir. Therefore, we analyzed the mRNA levels of TCF11/Nrf1 and its related transcription factor Nrf2. Interestingly, TCF11/Nrf1 mRNA levels significantly decreased upon nelfinavir treatment. On the other hand, Nrf2 mRNA levels increased when cells were treated with nelfinavir, particularly in OPM-2 cells. We next inspected levels of Nrf2-driven genes POMP and HMOX1. HMOX1 mRNA levels in particular significantly increased upon nelfinavir treatment compared to the DMSO-treated cells, suggesting that the Nrf2 protein level is also activated by nelfinavir (Figure 4).

### 2.5. Nelfinavir Sensitizes MM Cells to BTZ and CFZ Treatment

Nelfinavir has been reported to augment proteasome inhibition in myeloma cells by acting synergistically with BTZ, thus overcoming resistance to proteasome inhibitors [30]. This may be due to pan-proteasome activity, in which nelfinavir inhibits not only the β1/β5 active sites of the proteasome, but also the proteasomal β2 activity that is not targeted by proteasome inhibitors [24]. To inspect the effect of nelfinavir alone and in combination with BTZ and CFZ on proteasomal activity, we analyzed chymotrypsin-like activity of the proteasome in the OPM-2 and RPMI8226 MM cell lines (Figure 5). In both cell lines, nelfinavir significantly decreased proteasome activity compared to the cells treated with BTZ or CFZ alone. We next compared cytotoxic activity of NFV on MM cells in combination with BTZ or CFZ. 20 µM NFV is lethal to about 80% of the MM cells even when combined with sub-effective concentrations of BTZ or CFZ (5 nM). RPMI8226 cells were generally more sensitive to the co-treatment, for example, the combination of 10 µM NFV and 10 nM BTZ or CFZ was lethal to about 80% or 95%, respectively (Figure 6).

## 3. Discussion

In this study, we found that clinically used HIV PI nelfinavir inhibits TCF11/Nrf1-mediated proteasome re-synthesis by interfering with TCF11/Nrf1 proteolytic activation. The TCF11/Nrf1 pathway is an attractive therapeutic target for MM, because its activation upon proteasome impairment results in the synthesis of new proteasome subunits, thus augmenting the effect of proteasome inhibitors [5,11]. Previous studies implicated VCP/p97, NGLY1, and aspartyl protease DDI2 in TCF11/Nrf1 activation [6,7,10,31]. Recently, researchers found that inhibition of VCP/p97 or NGLY1 by small molecules potentiates the cytotoxic effect of carfilzomib in MM and the T cell-derived acute lymphoblastic leukemia [31]. Moreover, another recent study showed that this effect is recapitulated in the MDA-MB-231 cells that are DDI2-deficient [11]. These findings suggest that blocking the TCF11/Nrf1-mediated proteasome re-synthesis (or bounce-back response) can be advantageous in combination with proteasome inhibitors for treatment of hematological malignancies, and could also expand the use of proteasome inhibitors to solid tumors.

Previously, we found that the DDI2 protease domain (RVP) strikingly resembles the 3D structure of the HIV-1 protease (Figure 1B), but we were unable to detect binding of HIV PIs to DDI2 by a direct in vitro method employing recombinant DDI2 [12]. Previous studies have shown that several HIV PIs inhibit yeast Ddi1, Ddi1 from *Leishmania*, Ddi1 from *Plasmodium*, and human DDI1 in cell-based experiments [13,32]. The RVP domains of all these proteins are nearly identical to the human DDI2 RVP. The detailed mechanism of TCF/Nrf1 proteolytic activation by DDI2 is not yet understood, and it is unclear whether the substrate or protease needs to be post-translationally modified or whether another stimulus is needed to activate cleavage. During the revision of our manuscript, Yip and colleagues published an exciting article showing that yeast Ddi1 is a ubiquitin-dependent protease that cleaves only polyubiquitinated substrates [33]. It is very likely that human DDI2 might act in a similar way, although it still remains to be determined. Thus, we decided to evaluate the inhibitory activity of HIV PIs by cell-based methods reporting proteasome recovery and TCF11/Nrf1 transcriptional activity. Both reporter assays revealed similar inhibitory profiles, identifying nelfinavir as the most potent inhibitor and tipranavir as the second most potent. These results are in agreement with a previous study in which nelfinavir and tipranavir inhibited growth of a Ddi1-deficient yeast strain complemented with human DDI1, while other HIV PIs showed modest or no inhibition [13]. The IC_50_ and EC_50_ values calculated from dose responses of nelfinavir in both of our assays were in the low micromolar range and correlated with findings from the previous study [13]. Furthermore, the observed effective concentrations closely correspond to the plasma concentrations of nelfinavir observed in clinical trials [17,34]. Moreover, the resulting curves in both assays favored bi-sigmoidal fits, implying that nelfinavir has a dual mode of action in the range of concentrations used in this study.

Because nelfinavir significantly decreased protein levels of the TCF11/Nrf1 transcription factor in the nucleus, we assessed whether it directly affects TCF11/Nrf1 proteolytic activation. In all three cell lines tested, including MM cell lines, nelfinavir decreased the total mRNA and protein TCF11/Nrf1 levels upon activation of the pathway. Furthermore, nelfinavir significantly decreased levels of the processed (DDI2-activated) form of TCF11/Nrf1. Whether this is due to direct inhibition of DDI2 by nelfinavir still needs to be resolved. Interestingly, the mRNA expression levels of TCF11/Nrf1’s target proteasome subunits were not affected by nelfinavir, suggesting that the TCF11/Nrf1-related Nrf2 transcription factor likely stepped in, as nelfinavir is known to activate ER stress and oxidative stress [35,36,37]. Indeed, Nrf2 mRNA levels, as well as the genes driven by Nrf2 (POMP, HMOX1), were significantly upregulated upon nelfinavir treatment. We therefore speculate that in addition to crippling the TCF11/Nrf1 pathway, nelfinavir also activates proteasome subunit re-synthesis via Nrf2.

Taken together, our data and data from previous studies suggest that nelfinavir impedes TCF11/Nrf1-mediated proteasome re-synthesis at multiple sites (Figure 7). At high concentrations (≥40 μM), nelfinavir inhibits β1/β5 and β2 activity of the proteasome [24], thus activating TCF11/Nrf1-dependent proteasome re-synthesis (the bounce-back response). At the same time, nelfinavir inhibits UPR-activating protease S2P [19,20], which decreases active levels of the sterol regulatory element binding protein-1 (SREBP-1) transcription factor that is responsible for activating TCF11/Nrf1 transcription in the nucleus [38]. Therefore, TCF11/Nrf1 gene expression decreases. Furthermore, nelfinavir inhibits TCF11/Nrf1 proteolytic processing, resulting in lowered TCF11/Nrf1 protein levels in the nucleus. This decreases re-synthesis of proteasome genes by the TCF11/Nrf1 pathway. However, due to nelfinavir’s ability to activate ER and oxidative stress, proteasome re-synthesis might be, at least to some extent, re-driven by Nrf2. We inspected that further and, as shown in Appendix A, knockdown of NRF2 has a tendency to decrease mRNA levels of the analyzed proteasome genes. Further studies are needed to explain this phenomenon. Regardless of this opposing effect, overall proteasome activity is significantly decreased in the nelfinavir-treated MM cells, and nelfinavir potentiates BTZ activity. This is reflected in the overall cytotoxic effect of nelfinavir in combination with BTZ on myeloma cells, which is in agreement with findings from previous studies, including clinical trials [17,18,24,34,39]. Moreover, the observed ability of nelfinavir to activate the Nrf2 pathway suggests a possible mechanism for MM cell adaptation to nelfinavir treatment and a hypothetical direction for further improvements in MM therapy. Overall, the data presented here bring new insights into nelfinavir’s multiple activities in cancer cells and identify nelfinavir as the first inhibitor to inhibit TCF11/Nrf1 proteolytic activation.

## 4. Materials and Methods

### 4.1. Constructs

To obtain reporter plasmid 3xPSMA4-ARE-Luc, three copies of the ARE sequence (TGACTCTGCA) from the promoter region of the human PSMA4 gene were inserted into the pGL4.37 Vector (Promega, Madison, WI, USA) using XhoI and SacI restriction enzymes, as previously described [5]. Co-reporter plasmid encoding renilla luciferase pRL-TK (E2241) was purchased from Promega (Madison, WI, USA). Constructs pCMV-dR8.2 (Addgene, #8455) and pCMV-VSV-G (Addgene, #8454) required for lentiviral transduction were a kind gift from Bob Weinberg (MIT, Boston, MA, USA). Plasmid Ub^G76V^-GFP (Addgene, #11941) was a kind gift from Nico Dantuma, and construct pLV hU6-sgRNA hUbC-dCas9-KRAB-T2a-GFP (Addgene, #71237) was a kind gift from Charles Gersbach.

### 4.2. Cells and Inhibitors

The HEK293T and U2OS cell lines were grown in the DMEM containing 10% FBS and 2 mM L-glutamine (Biosera, Kansas City, MO, USA). The HEK293 cell line was grown in the IMDM containing 10% FBS and 2 mM L-glutamine. Human MM lines OPM-2 and RPMI8226 purchased from the Leibniz Institute DSMZ (Germany) were grown in RPMI 1640 (Biosera) supplemented with 10% FBS, 2 mM L-glutamine, and 2 ng/mL IL-6 (Sino Biological, Wayne, PA, USA). All cell lines used in the study were tested for mycoplasma contamination. Cells were treated with bortezomib (BTZ, UBPBio, PS341), MG132 (Sigma, M7449), or carfilzomib (CFZ, UBPBio, PR171) at various concentrations as indicated. Lopinavir (LPV), darunavir (DRV), brecanavir (BCV), tipranavir (TPV), amprenavir (APV), nelfinavir (NFV), indinavir (IDV), and ritonavir (RTV) were kindly provided by Dr. J. Konvalinka (Prague, Czech Republic).

### 4.3. Lentiviral Transductions

For lentiviral production, HEK293 cells were transfected with lentiviral construct pLV hU6-sgRNA-hUbC-dCas9-KRAB-T2a-GFP, expressing sgRNAs targeting DDI2 (5′- GACTCACTGAGCGTGTGTGA-3′), along with helper plasmids. The medium was changed 24 h after the transfection, and the supernatant containing the lentivirus was collected 48 and 72 h after the transfection. The lentivirus-containing medium was precipitated with the polyethylene glycol (PEG) solution (final concentration 8% PEG, 300 mM NaCl) and incubated overnight at 4 °C. The pellet was then suspended in the PBS, and aliquots were stored in liquid nitrogen at −80 °C. RMPI8226 cells were incubated with lentivirus particles (VLPs) in the presence of 5 μg/mL polybrene (Merck, Darmstadt, Germany) for 48 h. Cells were sorted based on GFP fluorescence.

### 4.4. Luciferase Assays

HEK293 cells stably expressing 3xPSMA4-ARE-Luc reporter were co-treated for 16 h with HIV PIs and/or vehicle (DMSO) in the presence or absence of 1 μM MG132. The cells were lysed, and the dual luciferase assay was performed as previously described [40]. The firefly luciferase activity was normalized to renilla luciferase control pRL-TK (Promega, E2241).

### 4.5. N-end Rule GFP Reporter Assays

U2OS cells in 96-well plates stably expressing Ub^G76V^-GFP reporter were treated with 200 nM carfilzomib (CFZ) for 2 h. The cells were washed with the PBS and treated with HIV PIs at various concentrations as indicated. The GFP fluorescence was measured after 20 h (endpoints, HIV PIs screening) or at indicated time points (every 30 min for 30 h, NFV EC_50_). The data were normalized to the CFZ-treated cells or the cells treated with the vehicle. To more accurately follow the dose dependence, the time course of GFP fluorescence was acquired, and the maximal value was used for subsequent regression.

### 4.6. Quantitative Reverse Transcription PCR

RNA was isolated using an RNAeasy Plus Micro Kit (Qiagen, Germantown, MD, USA). cDNA was prepared using a TATAA GrandScript cDNA Supermix (TATAAbiocenter) and Quantitect Reverse Transcription Kit (Qiagen) according to the manufacturer’s recommendations. Quantitative RT-PCR was performed using a LightCycler 480 (Roche Life Science, Penzberg, Germany). Primers used in qPCRs are listed in Appendix A. The statistics were determined using the Student’s unpaired *t*-test; *p* values < 0.05 were considered statistically significant.

### 4.7. Immunoblot Analysis

Cells were lysed in the SDS buffer (60 mM Tris [pH 6.0], 1.8% SDS, and 0.3 mM mercaptoethanol). For DDI2 detection, a rabbit polyclonal antibody (Bethyl A304-630; 1:1000) was used. For TCF11/Nrf1 detection, rabbit monoclonal antibody D5B10 (Cell Signaling, #8052; 1:1000) was used. Tubulin was visualized with a mouse polyclonal antibody (Sigma, #T6199; 1:2000). Immunoblots were developed using near-infrared fluorophore-conjugated anti-mouse and anti-rabbit secondary antibodies (IRDye 680RD Goat anti-Mouse; #926-68070, 1:15,000 and IRDye 800CW Goat anti-Rabbit, #926-32211, 1:15,000, from LI-COR) and visualized on an LI-COR Odyssey CLx NIR fluorescence reader. The images were quantified using the Fiji software [41]. Results show one representative of at least three independent experiments.

### 4.8. Proteasome Activity Assay and Cell Viability Assay

The OPM-2 and RPMI8226 myeloma cell lines were co-treated with 5 nM BTZ (or 5 nM CFZ) and 10 or 20 μM NFV for 24 h. The cells were isolated, suspended in a lysis buffer (50 mM HEPES [pH 7.5], 5 mM EDTA, 150 mM NaCl, 2 mM ATP, 1% Triton) and centrifuged at 15,000 rpm for 15 min at 4 °C. The cell lysates (20 μg total protein) were pre-incubated with 200 μM Suc-Leu-Leu-Val-Tyr-AMC (Bachem, I1395) fluorogenic substrate in an assay buffer (50 mM Tris [pH 8.0], 10 mM MgCl_2_, 1 mM ATP, 1 mM DTT) at 37 °C. The chymotrypsin-like activity of the proteasome was measured at 360/460 nm. The linear part of the time course of fluorescence was fitted using a non-linear regression method with 1/Y^2 weighting. The statistics were determined using the Dunnett’s test.

The OPM-2 and RPMI8226 cells (5 × 10^3^) in 96-well plates were treated for 24 h with BTZ (or CFZ) and/or NFV at the concentrations indicated. Cell viability was measured using a CellTiter-Glo (Promega, Madison, WI, USA) viability assay.

### 4.9. High-Throughput Confocal Imaging

HEK293 cells in 96-well plates were co-treated with 200 nM BTZ and 10 or 20 μM NFV for 16 h. The cells were washed with the PBS, fixed by methanol for 10 min, then washed again and treated with 10% FBS for 30 min. For TCF11/Nrf1 detection, rabbit monoclonal antibody D5B10 (Cell Signaling, #8052; 1:50) was used. Tubulin was visualized using a mouse monoclonal antibody (ThermoFisher, #33-2000). Secondary antibodies conjugated with fluorophore Cy3 (Jackson, Donkey anti-rabbit; 1:200) and fluorescein (Jackson, Donkey anti-mouse; 1:200) were used. Hoechst 34580 (ThermoFisher, 1:2000) was used to stain cell nuclei. The images were acquired using a Carl Zeiss LSM780 confocal scanning microscope and processed with the CellProfiler software [29]. Cell nuclei were identified using the Hoechst 34580 fluorescent dye, and the corresponding fluorescence of TCF11/Nrf1 was integrated.

### 4.10. Statistical Evaluation and Regression

Statistical evaluation and graph plotting were performed with GraphPad Prism v.8 (GraphPad Software, La Jolla, CA, USA). Sigmoidal and bi-sigmoidal regressions were obtained using the following linearized GraphPad Prism embedded Equations (1) and (2):(1)$$y=bottom+\frac{top-bottom}{1+{\left(\frac{EC_{50}}{x}\right)}^{Hill}}$$
(2)$$y=bottom+frac\frac{top-bottom}{1+{\left(\frac{EC_{50\_1}}{x}\right)}^{Hill_{1}}}+\left(1-frac\right)\frac{top-bottom}{1+{\left(\frac{EC_{50\_2}}{x}\right)}^{Hill_{2}}}$$

## 5. Conclusions

In this report, we bring new insights into how HIV protease inhibitor nelfinavir affects the growth of MM cells. First, we demonstrate that downregulation of aspartic protease DDI2 in MM cells leads to a significant decrease in proteasome gene expression. We further identify nelfinavir as the most potent HIV protease inhibitor that inhibits proteasome recovery. Nelfinavir accomplishes this by affecting TCF11/Nrf1-mediated proteasome re-synthesis by a dual mode of action. In a detailed analysis, we show that nelfinavir inhibits proteolytic processing of transcription factor TCF11/Nrf1, suggesting that it might interfere with DDI2 activity. Finally, we propose an overall model of nelfinavir’s modes of action related to proteasome synthesis and activity.

## Figures and Tables

**Figure 1 cancers-12-01065-f001:**
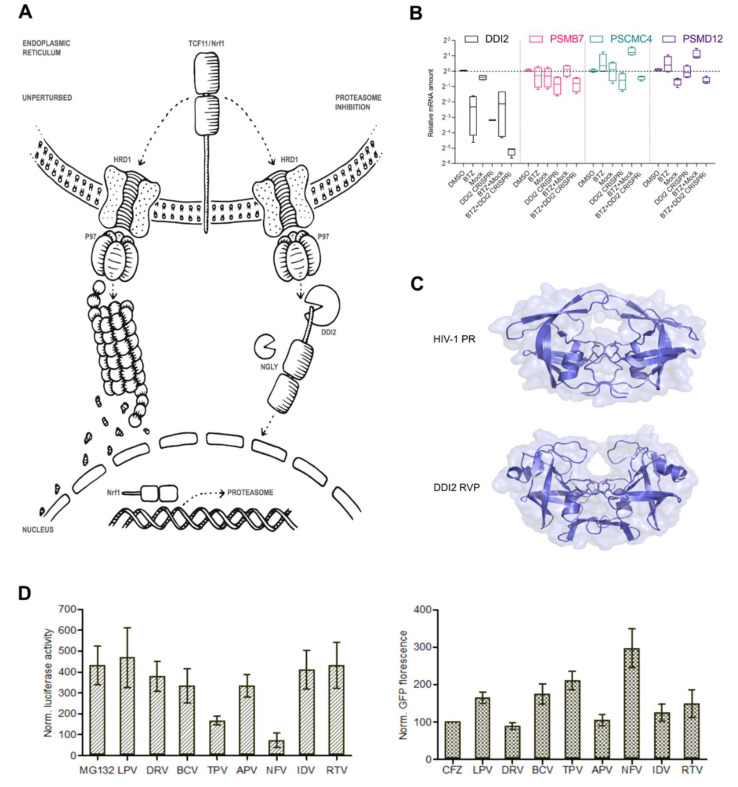
DDI2 is required for efficient proteasome re-synthesis in MM cells. (**A**) Current model of DDI2 function in the regulation of TCF11/Nrf1 transcriptional activity. Under normal conditions, TCF11/Nrf1 resides in the ER membrane, undergoes ubiquitination by HRD1, and is retrotranslocated by VCP/p97 to the cytosol, where it is degraded by the proteasome. When proteasome function is impaired, TCF11/Nrf1 is deglycosylated by N-glycanase 1 (NGLY1), extracted from the ER membrane by VCP/p97, and subsequently cleaved by DDI2. Active TCF11/Nrf1 translocates to the nucleus, binds to AREs, and activates proteasome gene expression. (**B**) Downregulation of DDI2 inhibits proteasome re-synthesis. RPMI8226 myeloma cells were treated with DDI2 CRISPRi lentiviral particles and BTZ (DDI2 CRISPRi + BTZ) for 16 h and compared to mock lentiviral particles and the BTZ control (BTZ + mock). Quantitative RT-qPCR with primers for the indicated genes was used to analyze the levels of DDI2 and proteasome gene expression. mRNA levels of GAPDH were used for normalization. The boxes show interquartile ranges, while whiskers denote minimal and maximal values (*n* = 4). (**C**) X-ray structure of the HIV-1 protease in an open conformation (Protein Data Bank code: 2pC0) [28] and X-ray structure of the retroviral protease domain (RVP) of DDI2 (Protein Data Bank code: 4rgh) [12]. (**D**) Left: Screening of HIV PIs used in clinical practice using a luciferase assay reporting TCF11/Nrf1 transcriptional activity. HEK293 cells stably expressing 3xPSMA4-ARE-Luc reporter were transfected with the renilla luciferase gene for normalization and co-treated with 1 μM MG132 and 10 μM HIV PIs. At 16 h post-transfection, a dual luciferase assay was used to measure luciferase activity. Normalized luciferase activity is shown. Error bars denote the SEM (*n* = 3). Right: Screening of HIV PIs with an N-end rule GFP reporter assay to measure proteasome activity. U2OS cells stably expressing Ub^G76V^-GFP reporter were treated with 200 nM CFZ for 2 h. The cells were washed with the PBS and treated with HIV PIs at 10 μM. The GFP fluorescence (dependent on proteasome activity) was measured 24 h after HIV PI treatment and normalized to the CFZ-treated cells.

**Figure 2 cancers-12-01065-f002:**
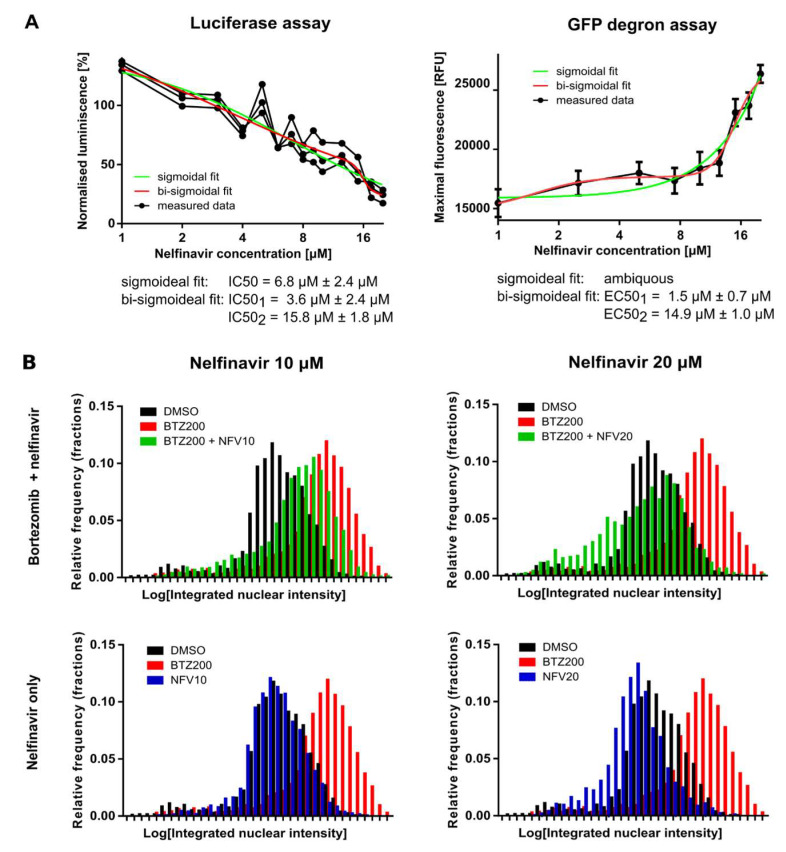
Nelfinavir inhibits Nrf1 transcriptional activity by a dual mode of action. (**A**) Left: HEK293 cells stably expressing 3×PSMA4-ARE-Luc reporter were transfected with renilla luciferase gene for normalization and co-treated with 1 μM MG132 and nelfinavir concentrations spanning 1 μM to 20 μM. At 16 h post-transfection, dual luciferase assays were used to measure firefly and renilla luciferase activity. Normalized luciferase activity is shown. The data were fitted by sigmoidal and bi-sigmoidal functions, and IC_50_ values were calculated. Right: U2OS cells stably expressing Ub^G76V^-GFP reporter were treated with nelfinavir at concentrations spanning 1 μM to 20 μM. The GFP fluorescence (dependent on proteasome activity) was measured at different time points (0–24 h), and maxima were measured. The data were fitted by sigmoidal and bi-sigmoidal functions, and EC_50_ values were calculated for the bi-sigmoidal fit. (**B**) HEK293 cells were co-treated with 200 nM BTZ and 10 μM or 20 μM NFV or DMSO for 16 h. TCF11/Nrf1 was detected using monoclonal antibody D5B10 (Cell Signaling, #8052; 1:50). The amount of nuclear TCF11/Nrf1 was estimated as integrated fluorescent nuclear intensity obtained by CellProfiler processing [29].

**Figure 3 cancers-12-01065-f003:**
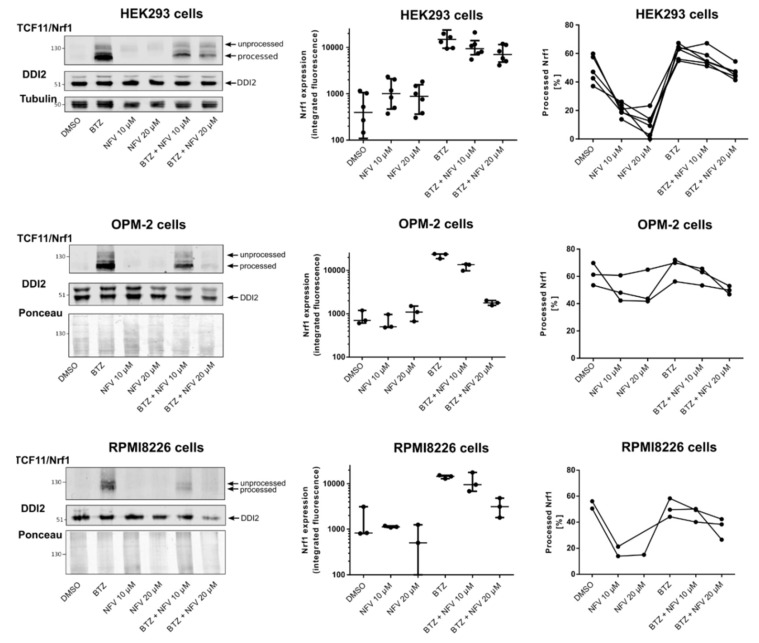
Nelfinavir inhibits TCF11/Nrf1 proteolytic processing. HEK293 cells and OPM-2 and RPMI8226 MM cells were treated with 10 μM or 20 μM nelfinavir alone or in combination with 200 nM BTZ (HEK293 cells) or 20 nM BTZ (OPM-2, RMPI8226 cells) for 16 h. Cells were lysed and subjected to immunoblot analysis visualizing TCF11/Nrf1, DDI2, and tubulin as the loading control for HEK293 cells. Ponceau staining was used as a loading control for myeloma cells. Total Nrf1 expression was quantified from the integrated fluorescence of corresponding TCF11/Nrf1 bands normalized to tubulin or the Ponceau loading control. The processed (activated) form of Nrf1 was quantified as shown as [%]. Data from six (HEK293 cells) and three (OPM-2, RPMI8226 cells) independent experiments are shown.

**Figure 4 cancers-12-01065-f004:**
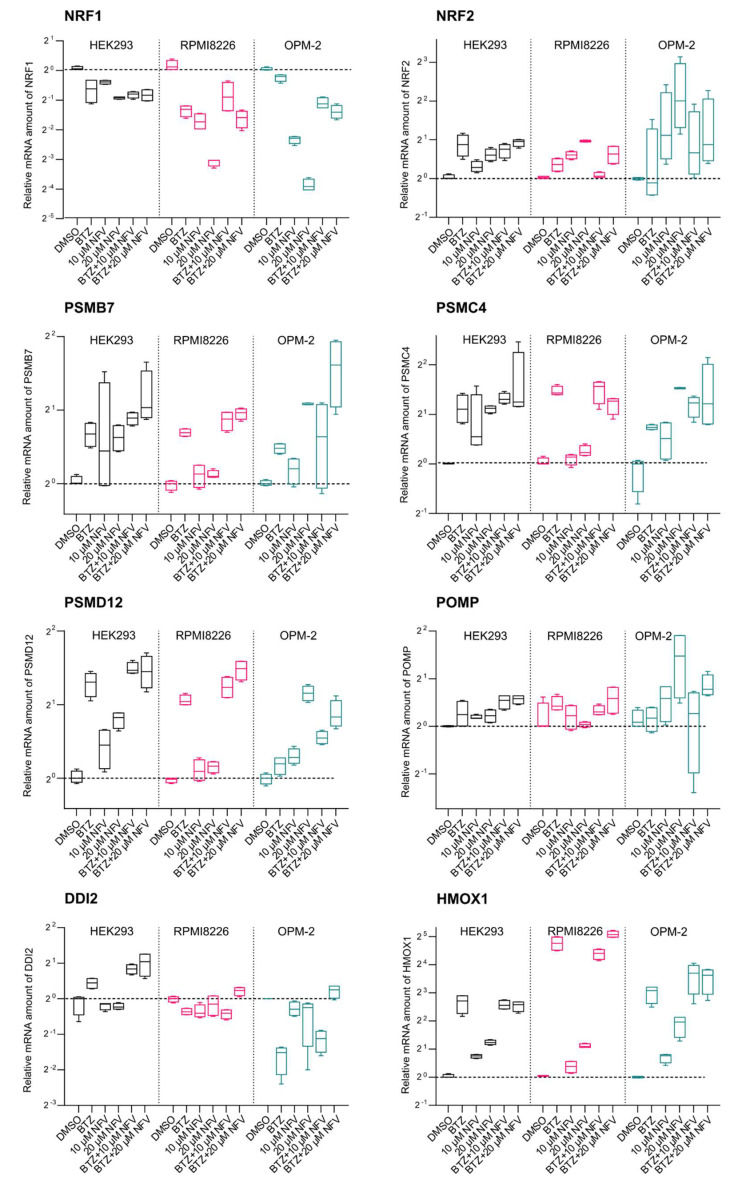
Nelfinavir decreases TCF11/Nrf1 gene expression and activates Nrf2. Quantitative RT-PCR analyses of the indicated genes are shown. HEK293, RPMI8226, and OPM-2 cells were treated with 200 nM BTZ (HEK293 cells) or 20 nM BTZ (RPMI8226, OPM-2 cells), 10 μM or 20 μM NFV, or a combination for 16 h. RNA extracted from the cells was converted to cDNA and used for RT-qPCR with the primers listed in Appendix A. mRNA levels of GAPDH were used for normalization. The boxes indicate interquartile ranges, while whiskers denote minimal and maximal values (*n* = 4).

**Figure 5 cancers-12-01065-f005:**
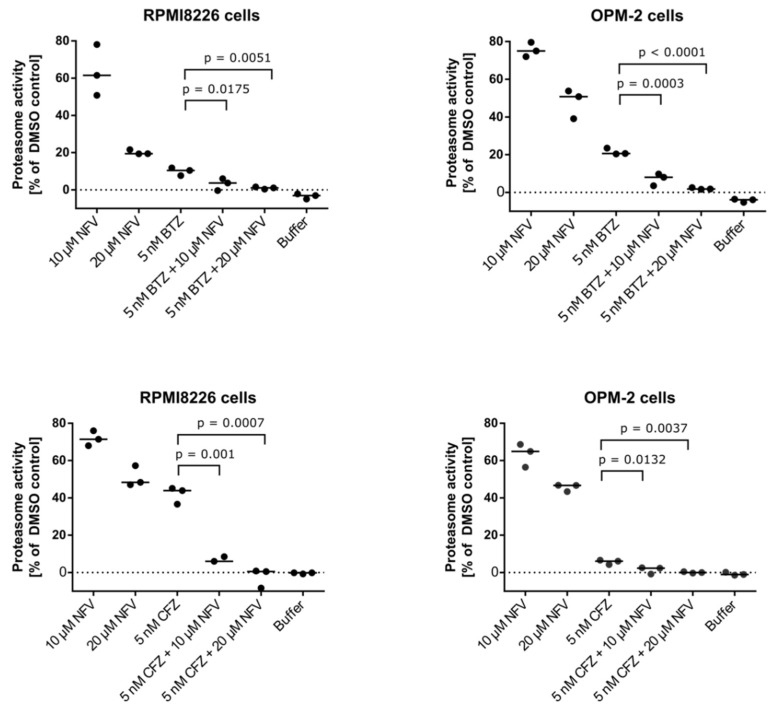
Nelfinavir potentiates activity of proteasome inhibitors. OPM-2 and RPMI8226 MM cell lines were co-treated with 5 nM BTZ or 5 nM CFZ and 10 μM or 20 μM nelfinavir for 24 h. Chymotrypsin-like activity of the proteasome was measured in cell lysates at 37 °C. To assess the level of significance, Dunnett’s multiple comparison tests were used, *p* values are indicated (*n* = 3).

**Figure 6 cancers-12-01065-f006:**
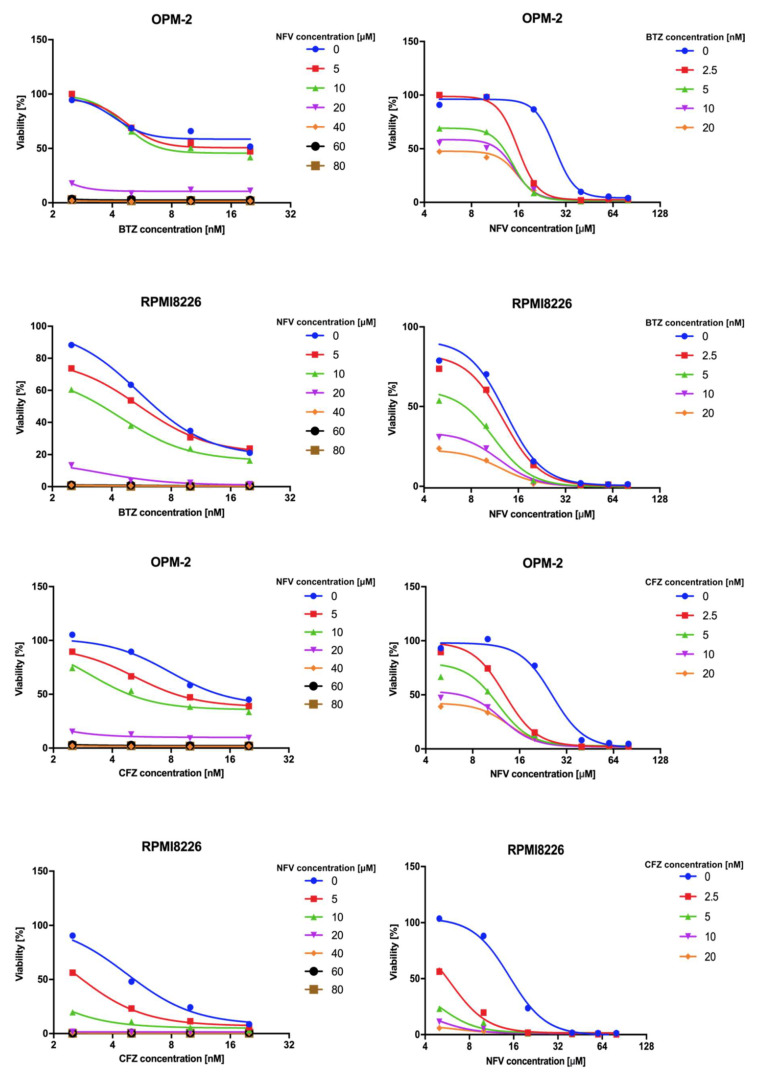
Cytotoxic activity of nelfinavir on OPM-2 and RPMI8226 cell lines. Both cell lines were incubated with increasing concentrations of BTZ or CFZ and nelfinavir as indicated, and cell viability was measured after 24 h with a CellTiter-Glo (Promega) viability assay. Data are shown as the median (*n* = 4).

**Figure 7 cancers-12-01065-f007:**
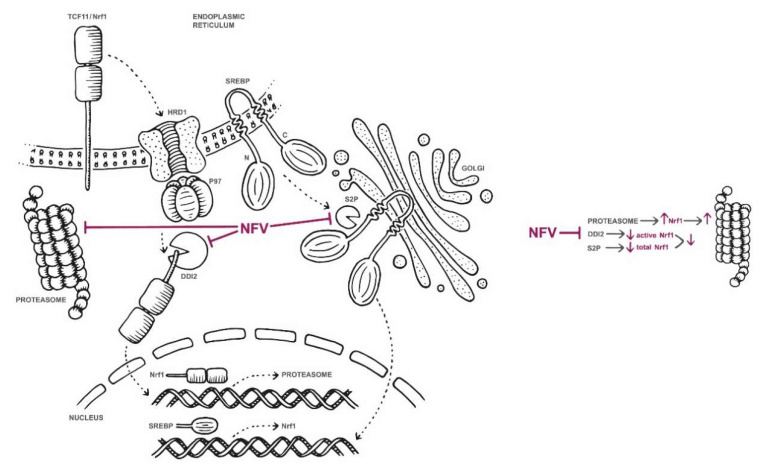
Suggested model of nelfinavir’s modes of action related to proteasome synthesis and activity. At concentrations ≥40 μM, nelfinavir partially inhibits β1/β5 and β2 activity of the proteasome [24], thus activating TCF11/Nrf1-dependent proteasome re-synthesis (the bounce-back response). At the same time, nelfinavir inhibits UPR-activating protease S2P [19,20], which decreases active levels of the SREBP-1 transcription factor that is responsible for activating TCF11/Nrf1 transcription in the nucleus [38]. Therefore, TCF11/Nrf1 gene expression decreases. Furthermore, nelfinavir inhibits TCF11/Nrf1 proteolytic processing, resulting in lowered TCF11/Nrf1 protein levels in the nucleus. This decreases re-synthesis of proteasome genes. However, due to nelfinavir’s ability to activate ER and oxidative stress, proteasome re-synthesis might be, at least to some extent, re-driven by the Nrf2 pathway.

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
