# Peer review of "Nelfinavir Inhibits the TCF11/Nrf1-Mediated Proteasome Recovery Pathway in Multiple Myeloma"

_cancers, 2020, doi:10.3390/cancers12051065_

Round 1

Reviewer 1 Report

The authors adequately modified the manuscript according to the reviewers' comments.

Reviewer 2 Report

The authors have answered and modified the article with sound reasons and novelty. Therefore, I recommend it to be published in the journal.

Reviewer 3 Report

The authors revised the manuscript appropriately according to the reviewers' comments.

This manuscript is a resubmission of an earlier submission. The following is a list of the peer review reports and author responses from that submission.

Round 1

Reviewer 1 Report

This manuscript describes the novel mechanism of nelfinavir on multiple myeloma cell lines in vitro. The authors found that nelfinavir had the most potent activity for downregulation of DDI2 and inhibition of the TCF11/Nfr1-pathway, which suppressed the bounce-back response to proteasome inhibition. They also showed that expression of TCF11/Nrf1 was inhibited by nelfinavir at both mRNA and protein levels in 2 myeloma cell lines. Moreover, nelfinavir enhanced cell death of these cell lines induced by bortezomib. Based on these data, they proposed the novel mechanism of action of nelfinavir for proteasome inhibition in multiple myeloma.

  1. The synergistic effects of nelfinavir and bortezomib have been reported and should be referred (Cell Death Dis 3 (7), e353, 2012).
  2. The cytotoxic effect of combination with nelfinavir and bortezomib was not clear and does not seem to have a significant difference although **** was shown in Figure 5A (but **** was not shown in the figure legends). It is recommended to use lower concentration of bortezomib or bortezomib-resistant myeloma cell lines to show the therapeutic potential of nelfinavir for enhancing the effects of proteasome inhibitors and overcoming the resistance to proteasome inhibition.
  3. It is recommended to use other proteasome inhibitors such as carfilzomib in combination with nelfinavir for cytotoxic assay in myeloma cells.
  4. The figure numbers (Figure 1D and Figure 3) should be included in the text.
  5. In Figure 6, Nrf2 pathway mentioned in the legends was not shown in the figure.

Author Response

Response to the review comments:

We would like to thank all three reviewers for their insightful comments and suggestions that have helped us to improve the manuscript. In the revised manuscript we indicated in “track-changes” mode all the changes that have been made and we specifically address the reviewer’s comments below.  

Reviewer 1:

Comments and Suggestions for Authors

This manuscript describes the novel mechanism of nelfinavir on multiple myeloma cell lines in vitro. The authors found that nelfinavir had the most potent activity for downregulation of DDI2 and inhibition of the TCF11/Nfr1-pathway, which suppressed the bounce-back response to proteasome inhibition. They also showed that expression of TCF11/Nrf1 was inhibited by nelfinavir at both mRNA and protein levels in 2 myeloma cell lines. Moreover, nelfinavir enhanced cell death of these cell lines induced by bortezomib. Based on these data, they proposed the novel mechanism of action of nelfinavir for proteasome inhibition in multiple myeloma.

  1. The synergistic effects of nelfinavir and bortezomib have been reported and should be referred (Cell Death Dis 3 (7), e353, 2012).

We have cited this work as reference # 30 in the original submission. However, we added the citation also on page 9, line 208, where it should be cited again. We thank the reviewer for pointing this out.

  1. The cytotoxic effect of combination with nelfinavir and bortezomib was not clear and does not seem to have a significant difference although **** was shown in Figure 5A (but **** was not shown in the figure legends). It is recommended to use lower concentration of bortezomib or bortezomib-resistant myeloma cell lines to show the therapeutic potential of nelfinavir for enhancing the effects of proteasome inhibitors and overcoming the resistance to proteasome inhibition.

We agree with the reviewer’s suggestion and therefore we re-tested the effect of NFV in combination with lower concentration of BTZ (5 nM) and we also prolonged the time of the treatment from 16h to 24h. Furthermore, we included CFV (5 nM) into the assay to have the analysis complete and concentrated only on MM cells. In the revised version there is a new Figure 5 with corrected Figure legend (page 9) and few changes in the text (page 9, lines 206, 211, 212-215). We are grateful for this particular suggestion since the new data considerably improved this particular analysis.

  1. It is recommended to use other proteasome inhibitors such as carfilzomib in combination with nelfinavir for cytotoxic assay in myeloma cells.

We did exactly as suggested and performed new cytotoxic assay also for CFZ. Moreover, we used a wider range of concentrations for BTZ, CFZ and NFV and prolonged the time of treatment from 16h to 24h. In the revised version there is a new Figure 6 with corrected Figure legend (page 10) and new commentary (page 9, lines 215-221). We are grateful for this particular suggestion since the new data also considerably improved this particular analysis.

  1. The figure numbers (Figure 1D and Figure 3) should be included in the text.

We thank the reviewer for pointing this out. Both figures are included in the revised version of the manuscript (Figure 1D, page 3, line 105 and Figure 3 on page 6, line 167).

  1. In Figure 6, Nrf2 pathway mentioned in the legends was not shown in the figure.

As suggested by reviewer #3, we performed a qPCR analysis to get more information on the mechanism of Nrf2 activation by NFV. We included the analysis in the Supplementary material as Figure 13S. The knockdown of NRF2 has a tendency to decrease mRNA levels of inspected proteasome genes, but the decrease is not significant compared to mock siRNA+NFV+BTZ. Further studies are needed to explain this phenomenon as we comment in the revised version on page 14, lines 295-298. We therefore revised our original working hypothesis to soften it up (page 14, line 316) and make it more inclusive regarding possible outcomes. Therefore, we suggest not to include Nrf2 pathway in Figure 7 (original Figure 6), if the reviewer agrees.  

Reviewer 2 Report

In this manuscript, Fassmannova et al. report the roles of inhibition of TCF/Nrf1 by Nelfinavir in myeloma.

This paper is well written and will provide several insights into the understanding of strategy for treatment of MM. However, several critical concerns are found in this article. The specific points are as follow.

(1) The molecular mechanism of acquired resistance to proteasome inhibitors through proteasome recovery by TCF/Nrf1 is not clear in this manuscript. So, I cannot find a significance of inhibition of TCF/Nrf1 by Nelfinavir in myeloma.

(2) The role of decrease of TCF/Nrf1 in the nucleus by Nelfinavir in anti-myeloma effect is not clear.

(3) Is it possible to determine whether TCF/Nrf1 is unprocessed or processed by Western blotting analysis?

(4) The description “nelfinavir increased BTZ’s proteasome inhibitory activity and significantly decreased proteasome activity compared to cells treated with BTZ alone.” is not consistent with that “MM cell lines appeared to be more sensitive to nelfinavir treatment in the absence of BTZ.”

Author Response

Response to the review comments:

We would like to thank all three reviewers for their insightful comments and suggestions that have helped us to improve the manuscript. In the revised manuscript we indicated in “track-changes” mode all the changes that have been made and we specifically address the reviewer’s comments below.  

Reviewer 2:

Comments and Suggestions for Authors

In this manuscript, Fassmannova et al. report the roles of inhibition of TCF/Nrf1 by Nelfinavir in myeloma.

This paper is well written and will provide several insights into the understanding of strategy for treatment of MM. However, several critical concerns are found in this article. The specific points are as follow.

(1) The molecular mechanism of acquired resistance to proteasome inhibitors through proteasome recovery by TCF/Nrf1 is not clear in this manuscript. So, I cannot find a significance of inhibition of TCF/Nrf1 by Nelfinavir in myeloma.

As we comment in the Introduction (page 1, lane 42 and 43 and Figure 1A), one important cause of resistance to proteasome inhibition is the so-called bounce-back response, in which proteasome activity is restored by TCF11/Nrf1-driven proteasome re-synthesis. In other words, when proteasome activity is impaired (for example by proteasome inhibitors), every cell (including MM cells) protects itself against this proteotoxic stress by inducing synthesis of the new proteasomes. This is exclusively driven by TCF11/Nrf1 transcription factor. By inhibiting Nrf1, or any key player involved in Nrf1 activation (e.i. NGLY, DDI2), one will potentiate the activity of proteasome inhibitors (and overcome resistance), as we show in our manuscript.

(2) The role of decrease of TCF/Nrf1 in the nucleus by Nelfinavir in anti-myeloma effect is not clear.

Nrf1 acts as a transcription factor, by decreasing its active levels in the nucleus one will impair proteasome re-synthesis (as mention above). This was not discovered by us, but by others we properly cite in the manuscript (e.i. ref #5-8, #11). We here show that NFV inhibits this pathway, potentiates proteasome inhibitors activity and thus we provide explanation, why NFV is effective in combination with BTZ in ongoing clinical trials with MM patients.

(3) Is it possible to determine whether TCF/Nrf1 is unprocessed or processed by Western blotting analysis?

Yes, it is. The antibody we used recognizes the full-length (unprocessed) and activated (processed) form of TCF11/Nrf1, as shown in Figure 3.

(4) The description “nelfinavir increased BTZ’s proteasome inhibitory activity and significantly decreased proteasome activity compared to cells treated with BTZ alone.” is not consistent with that “MM cell lines appeared to be more sensitive to nelfinavir treatment in the absence of BTZ.”

We thank the reviewer for pointing this out. As we re-tested the proteasome activities as suggested by reviewer #1, we deleted and reformulated the above mentioned sentences (page 9, lines 215-218).

Reviewer 3 Report

The authors used cell lines with HIV proteasome inhibitor showing that Nelfinavir inhibits the TCF11/Nrf1-mediated pathway in myeloma.

  1. In figure 2, it seems that the combination of Nelfinair with BTZ have effect on the NRF1 level, why the nelfinavir alone showed no difference with DMSO group?
  2. The Nelfinavir could activate NRF2 pathway, which might damper the effect of proteasome inhibition effect in Figure 4, could the authors add NRF2 inhibitor for more information of the real mechanism?

Author Response

Response to the review comments:

We would like to thank all three reviewers for their insightful comments and suggestions that have helped us to improve the manuscript. In the revised manuscript we indicated in “track-changes” mode all the changes that have been made and we specifically address the reviewer’s comments below.  

Reviewer 3:

Comments and Suggestions for Authors

The authors used cell lines with HIV proteasome inhibitor showing that Nelfinavir inhibits the TCF11/Nrf1-mediated pathway in myeloma.

  1. In Figure 2, it seems that the combination of Nelfinavir with BTZ have effect on the NRF1 level, why the nelfinavir alone showed no difference with DMSO group?

We thank the reviewer for pointing this out, since this is a very interesting observation. The possible explanation is that BTZ profoundly activates TCF11/Nrf1 signaling (approximately in one order of magnitude) expression, while we show that NFV has a dual inhibitory effect on TCF11/Nrf1 (on mRNA level by inhibiting S2P protease and on the protein level by inhibiting the processing of TCF11/Nrf1). However, in a physiological condition, the constitutive activity of TCF11/ Nrf1 is much lower, the presence of activated TCF11/Nrf1 transcription factor is in minute amounts and it is variable among individual cells. Therefore, it is difficult to follow smaller differences in TCF11/Nrf1 concentration in the nucleus. Despite that, we could observe a mild decrease in TCF11/Nrf1 concentration in the case of 20 µM nelfinavir (bottom right).

  1. The Nelfinavir could activate NRF2 pathway, which might damper the effect of proteasome inhibition effect in Figure 4, could the authors add NRF2 inhibitor for more information of the real mechanism?

Following the reviewer’s suggestion we have performed qPCR analysis and included the result of the experiment in the Supplementary material as Figure S13. We used siRNA targeting NRF2 instead of the NRF2 inhibitor to minimize the effects of possible “off-targets”. The knockdown of NRF2 has a tendency to decrease mRNA levels of inspected proteasome genes, but the decrease is not significant compared to mock siRNA+NFV+BTZ. Further studies are needed to explain this phenomenon as we comment in the revised version on page 14, lines 295-298. We therefore revised our original working hypothesis to soften it up (page 14, line 316) and make it more inclusive regarding possible outcomes.
